# Detection of Anti-Rift Valley Fever Virus Antibodies in Serum Samples of Patients with Suspected Arbovirus Infection

**DOI:** 10.3390/microorganisms11082081

**Published:** 2023-08-14

**Authors:** Daniele Lapa, Eliana Specchiarello, Massimo Francalancia, Enrico Girardi, Fabrizio Maggi, Anna Rosa Garbuglia

**Affiliations:** 1Laboratory of Virology, National Institute for Infectious Diseases Lazzaro Spallanzani (IRCCS), 00149 Rome, Italy; eliana.specchiarello@inmi.it (E.S.); massimo.francalancia@inmi.it (M.F.); fabrizio.maggi@inmi.it (F.M.); annarosa.garbuglia@inmi.it (A.R.G.); 2Scientific Direction, National Institute for Infectious Diseases Lazzaro Spallanzani (IRCCS), 00149 Rome, Italy; enrico.girardi@inmi.it

**Keywords:** Rift Valley virus, immunofluorescent assay, seroneutralization assay, arbovirosis

## Abstract

The definitive diagnosis of the Rift Valley fever virus (RVFV) requires a form of testing that is available only in reference laboratories. It includes indirect immunofluorescence assay (IFA), the serum neutralization assay (NA), and real-time PCR. Therefore, often, no attempts are made to detect it, even among travelers from endemic areas. In this study, the presence of anti-RVFV IgG and IgM was retrospectively screened in stored serum samples from people who were admitted with arbovirus symptoms at the National Institute for Infectious Diseases (INMI) L. Spallanzani, Rome, Italy. Overall, 80 residual serum samples were anonymized, and sub-aliquots were prepared and tested for anti-RVFV IgG and IgM. A serum neutralization assay was used as a confirmatory test. There was a positive result in eight out of 80 samples (10%) for anti-RVFV IgG, with titers ranging from 1:40 up to 1:1280. Three of eight (2.6%) samples were confirmed as seropositive through an in-house serum neutralization assay, with antibody titers ranging from 1:10 to 1:160. All samples resulted negative for anti-RVFV IgM and RVFV RNA when tested by IFA and real-time RT-PCR, respectively. Our data highlight that several RVFV infections can possibly escape routine virological diagnosis, which suggests RVFV testing should be set up in order to monitor virus prevalence.

## 1. Introduction

Rift Valley fever virus (RVFV) is one of eight pathogens (Ebola virus, Zika virus, Lassa fever virus, Nipah virus, Crimean–Congo hemorrhagic fever virus, severe acute respiratory syndrome coronavirus, and Middle East respiratory syndrome coronaviruses) included in the Bluepoint list by the World Health Organization (WHO) [1]. It is a mosquito-borne zoonotic viral disease that affects animals and humans and is transmitted mainly by the Aedes and Culex mosquito species. It is widespread, especially in South and Eastern Africa, Saudi Arabia, and Yemen [2]. RVFV is an RNA virus and has an incubation period of 2–6 days in humans [3]. People with RVFV usually have either no symptoms or a mild illness that includes fever, headache, weakness, back pain, vertigo, anorexia, photophobia, and dizziness [2,4]. However, 8–10% of people infected with RVFV develop severe symptoms, such as ocular disease (reported in 0.5–2% of patients), encephalitis, or inflammation of the brain (in less than 1% of patients), and hemorrhagic fever, which occurs in less than 1% of all RVF patients. Fatality for those people who do develop symptoms of hemorrhagic fever is around 50%, and death usually occurs 3–6 days after the onset of symptoms [5]. Several outbreaks have been described as causing severe economic and health consequences [1]. In humans, the virus can be detected in blood specimens up to days 4–5 post-onset of the symptoms by RT-PCR, antigen-capture assay, and/or viral isolation [6]. Specific anti-RVFV IgM antibodies appear on days 5–6 of symptoms, and afterwards, specific IgG antibodies can be detected and then persist for several years [6]. Due to known cross-reactions with other phleboviruses, serology results should be confirmed by a specific serum neutralization test [2]. Overall, 11 outbreaks of RVFV occurred between 2000 and 2016 in the Republic of Niger (2016), the Republic of Mauritania (2012), the Republic of South Africa (2010), Madagascar (2008 and 2009), Sudan (2007), Kenya, Somalia, Tanzania (2006), Egypt (2003), Saudi Arabia, and Yemen (2000) [4]. Nine hundred and fifty deaths were reported, with a fatality rate of 19.5% [4]. To date, no outbreaks have been reported in Europe [2]. There is a risk of arbovirus introduction to continents other than Africa, including Europe; the spread worldwide could be a result of a global distribution effect on arthropod vectors, considering that mosquitoes of the Culex and Aedes species are now circulating in Italy and Europe [4,7]. In Africa and Saudi Arabia, several studies in humans have been carried out on RVFV IgG prevalence. In humans, the percentage of seroprevalence ranges from 1.8% (Kenya) to 11.1% (Saudi Arabia) [8,9]. In countries surrounding the Mediterranean basin, anti-RVFV IgG prevalence ranged from 1.4% in Tunisia to 4.9% in Turkey [10,11]. Another important feature of this virus is its capacity of reassortment, and that it is a conserved event in the different RVFV strains [12,13]. The reassortment of RVFV with other closely related viruses is also a concern, especially with the co-circulation of multiple bunyavirus in the field. For example, Nigari virus was detected during an RVFV outbreak in Mauritania in 2010, evidencing a possible coinfection in a goat [14]. To date, in humans, no studies on RVFV IgG seroprevalence have been carried out in Europe.

The purpose of this investigation was to screen for the presence of IgG and IgM antibodies against RVFV in 80 randomly selected stored serum samples from people (37 females, 43 males; median age: 38 years; age range: 18–75 years) who were admitted with arbovirosis symptoms at the National Institute for Infectious Diseases (INMI) L. Spallanzani, Rome, Italy. The subjects had all come back from Africa or Middle East countries.

## 2. Materials and Methods

### 2.1. Sample Collection

All serum samples were anonymized, and sub-aliquots of each sample were prepared. We collected for this study 37 samples from females and 43 samples from males. These were evaluated for a panel of anti-arbovirus antibodies, including dengue virus, Zika virus, chikungunya virus, West Nile virus, Usutu virus, tick-borne-encephalitis virus, Japanese encephalitis virus, and phleboviruses by using specific commercial kits (Euroimmun, Lubecca, Germany). The presence of malaria infection was also evaluated by a rapid antigenic test (Careus malaria Rapydtest, Apacor, Seoul, Republic of Korea).

### 2.2. Virus Preparation

Initial passage, propagation, and titration were performed on Vero E6 cells (ATCC CRL-1586). The cells were maintained in minimal essential medium (MEM), containing 10% heat-inactivated fetal bovine serum (FBS) (Corning), L-glutamine (Corning), and penicillin/streptomycin solution (Corning).

Monolayers of VERO E6 cells were prepared 24 h before for RVFV propagation. The virus was obtained from the National Collection of Pathogenic Viruses (NCPV).

Confluent monolayer cells were washed with 1X PBS and then infected with RVFV at an MOI of 0.1 and incubated at 37 °C for 1 h. Finally, MEM 2% FBS was added and incubated at 37 °C. After 72 h, we observed a 90% cytopathic effect (CPE). The infected flask was frozen at −80 °C. The virus was then titrated.

Virus titer was determined by limiting dilution assay and residual infectivity was expressed as 50% Tissue Culture Infective Dose (TCID50/mL) calculated according to the Reed and Muench method. All work with infectious RVFV was performed under biosafety level 3 (BSL-3) conditions.

### 2.3. Preparation of Home-Made Slides

The slides with Vero E6 cells infected with RVFV at MOI 0.1 were home-made. The in-house slides were prepared using Vero E6 infected with RVFV; 24 h post-infection cells were trypsinized and mixed with uninfected Vero E6 in a proportion of 1:1, washed twice in Dulbecco’s phosphate-buffered saline 1× (Sigma-Aldrich, St. Louis, MO, USA) and fixed for 30 min in acetone at −80 °C. The acetone-fixed slides were allowed to dry under cabinet for several hours.

### 2.4. Indirect Immunofluorescent Assay

To detect anti-RVFV-IgM, serum samples were pre-treated with Eurosorb (Euroimmun, Lubecca, Germany) for 30 min at room temperature (RT), centrifuged at 3500 rpm for 10 min and diluted (screening dilution: 1:20) in normal saline (NS, Fresenius Kabi, Bad Homburg vor der Höhe, Germany) solution. For anti-RVFV-IgG tests, serum samples were directly diluted 1:20 in PBS1X. Each serum sample was serially diluted from 1:20 down to 1:1280 to estimate the antibody titer and incubated for 1 h at room temperature. The slides were washed with PBS 1X and incubated for 1 h at room temperature with anti-human IgM and IgG rabbit antibodies conjugated with FITC and counterstained with Evans Blue (Euroimmun, Lubecca, Germany). PBS-glycerol 1% was used as a mounting media (Euroimmun, Lubecca, Germany). The results were analyzed with a fluorescence microscope (Figure 1).

### 2.5. Neutralization Assay

Samples that resulted ≥1:20 by IFA were tested by an in-house serum neutralization assay in order to confirm this positivity.

Heat-inactivated and two-fold serial diluted sera were incubated at 37 °C 5% CO_2_ for 30 min with equal volumes of 100 Tissue Culture Infectious Dose (TCID50) RVFV. Then, 96-well tissue culture plates with sub-confluent Vero E6 cell monolayers were infected with 100 µL/well of virus-serum mixture and incubated at 37 °C and 5% CO_2_ for 30 min. Subsequently, the serum-virus was transferred onto the cells, and incubated a 37 °C. After 72 h, microplates were analyzed for the presence of the cytopathic effect (CPE).

### 2.6. Acid Nucleic Extraction and Real-Time PCR

Nucleic acids were extracted from all serum samples using QIAamp^®^ Viral RNA (QIAGEN, Hilden, Germany) according to manufacturing instructions. Briefly, we added 140 µL of serum to 560 µL buffer AVL containing carrier. After washing with AW1 and AW2 buffer, nucleic acids were eluted with 60 µL of AVE buffer. All extracted nucleic acids were tested in real-time RT-PCR (RealStar^®^ Rift Valley Fever Virus RT-PCR Kit 1.0) for RVFV RNA detection [15].

Amplification conditions were as follows: reverse-transcription 55 °C for 20 min, denaturation 95 °C for 2 min, then 45 cycles of 95 °C for 15 s, 55 °C for 45 s, 72 °C for 15 s [16]. This real-time RT-qPCR method did not cross-react with dengue virus, JEV, St. Louis encephalitis virus, Usutu virus, Marburg virus, Ebola virus, West Nile virus, yellow fever virus, nor Zika virus [17].

The analytical assay sensitivity was 0.89 copies/μL (95% confidence interval (CI): 0.52 to 2.09 copies/μL) [17].

### 2.7. Statistical Analysis

Statistical analysis was performed using SPSS version 21 software (SPSS Inc., Chicago, IL, USA). Categorical variables are described with absolute frequencies or percentage frequencies and continuous variables are expressed as mean values or median and range.

Inferential statistics to test differences in patients’ characteristics were assessed by Fisher’s exact test for categorical variables, and comparison of continuous variables were made using the Mann–Whitney test. A *p* value < 0.05 was considered to be statistically significant.

## 3. Results

Overall, 8 out of 80 serum samples (10%) were found to be anti-RVFV IgG positive by IFA assay, but only 3 out of 8 (2.6%) were confirmed by a serum-neutralization assay. Anti-RVFV IgG titers ranged from 1:40 to 1:1280 in IFA assay, while a lower titer was observed in the neutralization assay from 1:10 to 1:160. All samples were negative for anti-RVFV IgM and RVFV RNA.

Overall, 10 samples tested positive for anti-dengue virus IgG, and 6 were positive for an antigen of the malaria. The samples were also tested for anti-RVFV IgG and IgM using an in-house indirect immunofluorescence assay (IFA).

Among the 8 anti-RVFV IgG reactive sera, one serum, which was confirmed RVFV positive by neutralization assay, was also positive for malaria antigen. All the samples tested negative for the other pathogens linked to arbovirosis (Table 1). No cross-reactivity was found as the samples were tested for other arboviruses, which suggests a high specificity of the serology methods used. No statistically significant association was found with age. Negative anti-IgG-RVFV patients had a mean age of 41 y versus 40 y of anti-IgG-RVFV positive patients (*p* > 0.05).

## 4. Discussion

RVFV, being a zoonosis that affects not only humans but also livestock for breeding, has an important impact on public health and the economy in the regions where it occurs. For example, Saudi Arabia and Yemen suffered economic losses of 10M USD and 107M USD, respectively, during the last outbreaks [5].

Due to the presence of a wide range of host and vector species, RVFV is spreading also in non-endemic regions. Hence, the risk of RVFV introduction into Europe is high [4].

In Africa and Saudi Arabian countries, RVFV seroprevalence in RVFV-related arboviruses range 2.1–9% in humans [18,19,20,21].The distribution of Culex pipiens and Aedes albopictus is widespread in European countries and they are able to transmit the virus [22,23,24]. These mosquitoes are especially present in Albania, Croatia, France, Greece, Monaco, Montenegro, Italy, San Marino, Slovenia, and Spain [24].

Little information is available regarding the seroprevalence of anti-RVFV antibodies or autochthonous cases in non-endemic areas because RVFV diagnosis is still uncommon in those regions. Currently, no antibody surveys toward RVFV have been performed in human samples from European countries. The only seroprevalence study was carried out in Poland, with as many as 973 bovine serum samples being screened, and the results were negative for anti-RVFV IgG [25]. In the present study, anti-RVFV IgG antibodies were found in 2.6% (3/80) of a group of human blood samples stored at INMI L. Spallanzani, Rome. The absence of anti-RVFV IgM and viral RNA in these samples suggests that all IgG serostatus was linked to previous infections. Our data also indicate that several RVFV infections could escape routine virological diagnosis and that the seroprevalence of the virus could be also underestimated in non-endemic countries.

Moreover, our data show a discrepancy between the results obtained with the ELISA method and those obtained with serum neutralization. Indeed only 3/8 cases that tested positive with the ELISA method were confirmed in the serum neutralization assay. This may indicate a higher specificity of the seroneutralization assay or a lower sensitivity of this test. Seroneutralization requires high quality equipment and well-trained personnel, which are not always available in resource-limited countries, thus the ELISA method is a more versatile test than the seroneutralization assay. Nevertheless, ELISA tests show cross-reactivity with other viruses, such as Rio Grande virus [26]; therefore, ELISA-positive results should be confirmed with a second test based on different viral proteins.

A surveillance system could be useful to obtain an effective control of RVFV in non-endemic areas. It could be carried out by testing travelers mainly from Africa or Middle East countries [27]. Evidence of RVFV-positive individuals returning from trips to African countries has also been reported by ECDC [2]. However, a sentinel network is considered to be a very expensive system and, therefore, difficult to set up [28]. The other method that could limit a spread of the virus is rapid diagnosis in suspected cases. Recently a rapid real-time reverse transcriptase isothermal amplification (RT-LAMP) system was developed. It provides results in 30 min. Diagnostic sensitivity and specificity of this RT-LAMP reaches 98.36% and 96.49% in comparison with qRT-PCR, which was performed with a ten-fold seral dilution of the known concentration of RVFV total RNA with an initial concentration of 18.5 pg/µL until 1,850,000 pg. The RT-LAMP was found to be ten-fold more sensitive compared to the RVFV qRT-PCR assay [29].

Another assay was developed for the detection of nucleoprotein (N) of RVFV using the lateral flow immunochromatographic strip test (LFT) and the results were recorded after 15 min. The analytical assay sensitivity was 100% (CI 95% (90.1; 100)); while the analytical assay specificity was 98.81% (CI 95% (95.8; 99.7)) [30].

Given the parameters of this study, there were a number of limitations. Only samples previously stored at INMI L. Spallanzani were used, which limited our geographic approach, and they did not provide any seroprevalence information for the remainder of the country or regarding the geographic area where the patients had contracted the infection. The discrepancies between ELISA and seroneutralization assay suggest that an unequivocal diagnosis of RVFV is complicated. However, despite these difficulties, it is very important to develop new tests that will provide effective support in outbreak emergence management and the surveillance of possible RVFV spread in non-endemic areas.

## Figures and Tables

**Figure 1 microorganisms-11-02081-f001:**
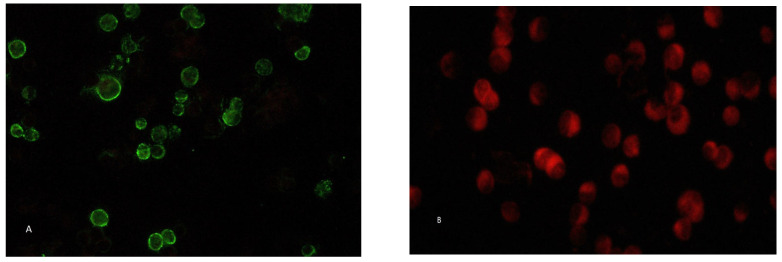
Indirect immunofluorescence assay with E6 Vero cells infected with RVFV at MOI 0.1 (**A**) E6 cells incubated with RVFV negative control provided by EUROIMMUN kit (**B**). Panel A shows a specific staining of RVFV infected cells when incubated with an ID40 serum sample (Dilution 1:40), whereas the negative RVFV serum tested negative. Fluorescent images were viewed with a Nikon Eclipse E600 20× Fluar lens, and digital images were taken with a Nikon DS F13 digital camera and Nikon Nis-Elements software v5.01.

**Table 1 microorganisms-11-02081-t001:** Neutralization results of samples that were anti-RVFV IgG positive with indirect immunofluorescence assay (IFA).

ID	Age (y)	Other Infections * Detected	RVFV Real-Time RT-PCR	Anti-RVFV IgM	Anti-RFVF IgG	Neutralization RVFV Assay
10	29	None	Undetected	<1:10	1:40	**1:10**
32	48	None	Undetected	<1:10	1:40	<1:10
34	27	None	Undetected	<1:10	1:40	<1:10
40	26	None	Undetected	<1:10	1:40	**1:10**
47	21	None	Undetected	<1:10	1:40	<1:10
55	62	None	Undetected	<1:10	1:40	<1:10
79	75	None	Undetected	<1:10	1:40	<1:10
80	30	Malaria	Undetected	<1:10	1: 1280	**1:160**

* These infections included: dengue, Zika, chikungunya, West Nile, Usutu, tick-borne encephalitis, Japanese encephalitis, phleboviruses, and malaria; y, years.

## Data Availability

The present study used anonymized samples that could not be traced back to patients in any way and, therefore, did not require any ethical approval.

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
