# Peer review of "Detection of Anti-Rift Valley Fever Virus Antibodies in Serum Samples of Patients with Suspected Arbovirus Infection"

_microorganisms, 2023, doi:10.3390/microorganisms11082081_

Round 1

Reviewer 1 Report

The submission overall is very well put together both grammatically and functionally.  The authors were able to perform a retrospective study on serum samples collected from travelers returning from Africa and identify RVFV in a subset of the samples. But without more details about the demographics and travel history of these travelers the data does not establish a clear conclusion.  If this data is available and can be added to the manuscript it might increase relevance of the 3 RFVF positives seen in the 80 serum samples.  Without it, we are only able to assess that the study found previous infection in a limited cohort with known travel to an endemic area for RFVF.  Something that would be expected.  As written this study does not provide sufficient information to draw a sold conclusion, and correclty the authors did not try to over analyze the data.  

Author Response

Referee #1

The submission overall is very well put together both grammatically and functionally.  The authors were able to perform a retrospective study on serum samples collected from travelers returning from Africa and identify RVFV in a subset of the samples. But without more details about the demographics and travel history of these travelers the data does not establish a clear conclusion.  If this data is available and can be added to the manuscript it might increase relevance of the 3 RFVF positives seen in the 80 serum samples.  Without it, we are only able to assess that the study found previous infection in a limited cohort with known travel to an endemic area for RFVF.  Something that would be expected.  As written this study does not provide sufficient information to draw a sold conclusion, and correclty the authors did not try to over analyze the data.  

Response: The referee is absolutely right. This study has limitations since it does not give any indications concerning the exact country where  the patients travelled. It  also included a limited number of samples. These limits are mentioned in the discussion (Lines 229-234). However, the aim of this work was to investigate the presence of anti-RVFV antibodies among subjects who had traveled abroad, mainly in Africa, but also in Middle Eastern countries such as Yemen, Saudi Arabia to verify the presence of antibodies against RVFV in humans. Sera, that  belonged to subjects who attended  our Medical Center for suspected arbovirus disease, were used. All of them had come back from areas where RVFV is circulating.

The serum samples, in compliance with the privacy regulations in force in our country, have been anonymised. For this reason the countries where the patients stayed are not indicated in the text. These data, even though they refer to a limited sample size, highlight a non-negligible prevalence of anti-RVFV antibodies in the study group. They also suggest the need to introduce RVFV testing  into the arbovirus diagnostic algorithm and to enhance the detection of this virus in order to improve general preparedness in the control of highly contagious infections (Lines 234-236) .

These data are to be considered preliminary and we have already started a memorandum of understanding with the Ministry of Health to carry out a seroprevalence study among migrants living in temporary refugee reception centre  and at a veterinary level among Culex in Italy and in subjects with arbovirus symptoms returning from countries where RVFV is endemic.

Reviewer 2 Report

Abstract

Line 15-22 adjust the font type and size

line 16 "Serum neutralization assay was used..."

Introduction

line 32 Culex and Aedes in italic form

line 33 Add reference(s) for this statement "It is diffused, especially in South and Eastern Africa, Saudi Arabia, and Yemen"

line 34 Add reference(s) for this statement "RVFV is an RNA virus and has an incubation period of 2–6 days in humans"

line 46 Known, K should not be capitalized

Method

line 75 Tick-borne encephalitis not tick-born-

line 76 "Phleboviruses" not Pheboviruses

line 76 Which phlebovirus kits did the authors used from Euroimmun, Germany? Was it a specific virus or universal Phleboviruses?

line 104 fluorescent

line 119-124 adjust font type and size

line 128 "After we added 140 µL, serum to the Buffer AVL–carrier RNA in the microcentrifuge tube." please clarify the sentence

line 126-148 authors may shorten the protocol briefly while keeping some volumes added which are different from the protocol given by the manufacturer

line 150 "This method of PCR no cross-react with Dengue virus" modify to "This real-time RT-qPCR method did not cross-react with Dengue virus"

line 150-151 "This method of PCR no cross-react with Dengue virus" please add reference for this statement

Results

line 172 "No statistically significant association was found with age and

sex (p>0.05)." However, there is no data regarding human patients (in a table form is better).

line 176 table legend "Chikungunya" "Tick-borne encephalitis" "Phleboviruses"

table 1 adjust the font type in the table

Can authors present image results of "Indirect fluorescent assay visualized with the fluorescence microscope"?

I am wondering why there is no data on the samples, e.g. year collected, local residents or traveler, etc. Please give additional information or explanation if such data is confidential.

Discussion

line 182 In fact

line 184 consider modifying "resulted in less food availability" to "resulted in food scarcity"

line 186 earnings

line 188 consider modifying "like" to "such as"

line 189-195 The minimum number of sentences required to create a complete paragraph is 3. Please revise.

line 199-200 italicize species names

line 179-204 These are already mentioned in the introduction. Authors may mention a bit of introduction in the Discussion section very briefly. Please shorten to one or two paragraphs at most. And later focus on discussing the results obtained.

line 225 ten-fold

line 231. Please also discuss the method used in this study, advantages and disadvantages, and whether this method could be useful for future screenings.

line 233 "Only in Poland a seroprevalence study was carried out on 973 bovine serum samples. All samples resulted negative for anti- RVFV IgG" consider modify into

"The only seroprevalence study was carried out in Poland, screening as many as 973 bovine serum samples, and the results were negative for anti-RVFV IgG"

line 235-239

Please elaborate more on the migration route of the travelers, any data mentioning the flow, such as the origin country and destination in the Europe? Any study on the arboviral-diseases spread from Middle-east/Africa to Europe? Regarding the data on this study, 80 serum, is it related to any travelers?

line 235

"Considering these preliminary data, still lacking seroprevalence studies of migrants from geographic areas of Africa" --> this study is not about Africa

line 238

"continuous migration flows there is an increased risk of RVFV introduction" any prior studies to justify? or only inferred by authors? "continuous migration flows MAY increase the risk of RVFV introduction" if there is no justification.

line 240-242

Please explain how the results of this study, i.e. RVFV infection in a population in Italy, impact the society, the country, and neighboring area. What are the consequences, future screenings, etc?

References

In general, please VERIFY all of the cited references.

For example, the reference no. 14; the format is not complete (authors of the article, when the article was written, accessed date, and importantly, the link was broken).

Some sentences are difficult to comprehend, particularly in the Discussion section.

Author Response

Referee#2

Comments and Suggestions for Authors

Abstract

Line 15-22 adjust the font type and size

Response: done

line 16 "Serum neutralization assay was used..."

Response: done

Introduction

line 32 Culex and Aedes in italic form

Response: done

line 33 Add reference(s) for this statement "It is diffused, especially in South and Eastern Africa, Saudi Arabia, and Yemen"

Response: We have added this reference 2

line 34 Add reference(s) for this statement "RVFV is an RNA virus and has an incubation period of 2–6 days in humans"

Response: We have added this reference 3

line 46 Known, K should not be capitalized

Response: done

Method

line 75 Tick-borne encephalitis not tick-born-

Response: done

line 76 "Phleboviruses" not Pheboviruses

Response: done

line 76 Which phlebovirus kits did the authors used from Euroimmun, Germany? Was it a specific virus or universal Phleboviruses?

Response: The kit we  used was able to identify  Napoli, Sicilia, Cyprus, Toscana strains.

line 104 fluorescent

Response:done

line 119-124 adjust font type and size

Response: done

line 128 "After we added 140 µL, serum to the Buffer AVL–carrier RNA in the microcentrifuge tube." please clarify the sentence

Response: we have modified this paragraph

line 126-148 authors may shorten the protocol briefly while keeping some volumes added which are different from the protocol given by the manufacturer

Response: We resumed this description as suggested by the referee (Lines 126-131).

line 150 "This method of PCR no cross-react with Dengue virus" modify to "This real-time RT-qPCR method did not cross-react with Dengue virus"

Response: done

line 150-151 "This method of PCR no cross-react with Dengue virus" please add reference for this statement

Response: We have added a reference  ( Ref 17)

Results

line 172 "No statistically significant association was found with age and

sex (p>0.05)." However, there is no data regarding human patients (in a table form is better).

Response: we have added some data ( line 161-162 ) and we have added the age in Table 1

line 176 table legend "Chikungunya" "Tick-borne encephalitis" "Phleboviruses"

Response: done

Table 1 adjust the font type in the table

Response: done

Can authors present image results of "Indirect fluorescent assay visualized with the fluorescence microscope"?

Response:We have added Figure1

I am wondering why there is no data on the samples, e.g. year collected, local residents or traveler, etc. Please give additional information or explanation if such data is confidential.

 Response: The serum samples were anonymized according to Italian national policy on privacy rights.

Discussion

line 182 In fact

Response: It has been removed from the text

line 184 consider modifying "resulted in less food availability" to "resulted in food scarcity"

Response: It has been removed from the text

line 186 earnings

Response: It has been removed from the text

line 188 consider modifying "like" to "such as"

Response: It has been removed from the text

line 189-195 The minimum number of sentences required to create a complete paragraph is 3. Please revise.

Response: We have modified this paragraph

line 199-200 italicize species names

Response: done

line 179-204 These are already mentioned in the introduction. Authors may mention a bit of introduction in the Discussion section very briefly. Please shorten to one or two paragraphs at most. And later focus on discussing the results obtained.

Response: We have modified this part of the discussion according to the reviewer’s suggestion.

line 225 ten-fold

Response: done

line 231. Please also discuss the method used in this study, advantages and disadvantages, and whether this method could be useful for future screenings.

Response: We have added some sentences concerning this issue ( Lines 204-212).

line 233 "Only in Poland a seroprevalence study was carried out on 973 bovine serum samples. All samples resulted negative for anti- RVFV IgG" consider modify into

"The only seroprevalence study was carried out in Poland, screening as many as 973 bovine serum samples, and the results were negative for anti-RVFV IgG"

Response: done (Lines 196-198)

line 235-239

Please elaborate more on the migration route of the travelers, any data mentioning the flow, such as the origin country and destination in the Europe? Any study on the arboviral-diseases spread from Middle-east/Africa to Europe? Regarding the data on this study, 80 serum, is it related to any travelers?

Response: This part had been removed

line 235

"Considering these preliminary data, still lacking seroprevalence studies of migrants from geographic areas of Africa" --> this study is not about Africa

Response: This part has been removed

line 238

"continuous migration flows there is an increased risk of RVFV introduction" any prior studies to justify? or only inferred by authors? "continuous migration flows MAY increase the risk of RVFV introduction" if there is no justification.

Response: This part has been removed

line 240-242

Please explain how the results of this study, i.e. RVFV infection in a population in Italy, impact the society, the country, and neighboring area. What are the consequences, future screenings, etc?

 Response: We have added some sentences (Lines 229-234)

References

In general, please VERIFY all of the cited references.

For example, the reference no. 14; the format is not complete (authors of the article, when the article was written, accessed date, and importantly, the link was broken).

Response: done

Comments on the Quality of English Language

Some sentences are difficult to comprehend, particularly in the Discussion section.

Response: The English had been revised by an English tongue Professor

Reviewer 3 Report

The manuscript is about detecting anti-Rift Valley fever virus antibodies in patients with suspected arbovirus infection.

Introduction

It is ok.

Materials and Methods

There is no ethical approval to publish this study. 

Please provide details on how samples were collected and stored. How many samples were collected from males and females? 

Line 78-79: "Overall, 10 samples tested 78 positive for anti-Dengue virus IgG, and 6 were positive for antigen of the malaria." These are the results.

There are no references in methods, please add references.

Discussion

Line 199-200: Italicize the scientific names of mosquitoes.

Some paragraphs have only one sentence. You can merge these paragraphs.

Conclusion

There is no separate conclusion. Please provide a conclusion. 

References

References are poorly written, e.g. 2, 4, 5, 14, 23. There is no data retrieval date. 

Moderate editing of the English language is required.

Author Response

Referee#3

The manuscript is about detecting anti-Rift Valley fever virus antibodies in patients with suspected arbovirus infection.

Introduction

It is ok.

Materials and Methods

There is no ethical approval to publish this study.

Response. The samples were anonymized thus no Ethical approval is required ( lines 248-249 )

Please provide details on how samples were collected and stored. How many samples were collected from males and females?

Response . These aspects are described in line 76

Line 78-79: "Overall, 10 samples tested 78 positive for anti-Dengue virus IgG, and 6 were positive for antigen of the malaria." These are the results.

Response. Thank you for this suggestion. We included this information in Results section ( lines 153-155)

There are no references in methods, please add references.

Response: Two References had ben added(Ref 15 and 16)

Discussion

Line 199-200: Italicize the scientific names of mosquitoes.

Response. done

Some paragraphs have only one sentence. You can merge these paragraphs.

Response. Done

Conclusion

There is no separate conclusion. Please provide a conclusion.

Response: We modified the discussion and we added a conclusion ( Lines 234-236)

References

References are poorly written, e.g. 2, 4, 5, 14, 23. There is no data retrieval date.

Response: Thank you for this observation. We checked the reference

Round 2

Reviewer 2 Report

The authors have addressed all comments and the manuscript has also generally improved.

Please follow the MDPI Reference format. Particularly when using Website and Online Resources.

Format:

Author (if available). Title of the webpage (if available). Available online: http://... (accessed on date).

Example:

International Union of Pure and Applied Chemistry Home Page. Available online: http://www.iupac.org/dhtml_home.html (accessed on 24 April 2005).

Author Response

Referee#2

Please follow the MDPI Reference format. Particularly when using Website and Online Resources.

Response: we have modified the References. All reviewer comments have been highlighted in green

Reviewer 3 Report

Authors complied with all comments.

Please proofread manuscript carefully.

Author Response

Referee#3

Please proofread manuscript carefully.

Response: done. Full manuscript text has been revised by a professional Mother-tongue scientific English editor, Dr.            Michael Kenyon

All reviewer comments have been highlighted in green
